# Impact of the Covid-19 pandemic on audiology service delivery: Observational study of the role of social media in patient communication

**Adeel Hussain**[1], **Zain Hussain**[2]*, **Mandar Gogate**[1], **Kia Dashtipour**[1], **Dominic Ng**[2], **Muhammed Shaan Riaz**[3], **Adele Goman**[1], **Aziz Sheikh**[4], **Amir Hussain**[1]

1 School of Computing, Edinburgh Napier University, Edinburgh, Scotland, 2 Edinburgh Medical School, Chancellor's Building, The University of Edinburgh, Edinburgh, Scotland, 3 Gloucestershire Hospitals NHS Trust, England, United Kingdom, 4 Usher Institute, The University of Edinburgh, Edinburgh, Scotland

* zain.hussain@ed.ac.uk

**Data Availability Statement:** The data used in this study is now available on GitHub at [https://github.

## Abstract

The Covid-19 pandemic has highlighted an era in hearing health care that necessitates a comprehensive rethinking of audiology service delivery. There has been a significant increase in the number of individuals with hearing loss who seek information online. An estimated 430 million individuals worldwide suffer from hearing loss, including 11 million in the United Kingdom. The objective of this study was to identify National Health Service (NHS) audiology service social media posts and understand how they were used to communicate service changes within audiology departments at the onset of the Covid-19 pandemic. Facebook and Twitter posts relating to audiology were extracted over a six week period (March 23 to April 30 2020) from the United Kingdom. We manually filtered the posts to remove those not directly linked to NHS audiology service communication. The extracted data was then geospatially mapped, and themes of interest were identified via a manual review. We also calculated interactions (likes, shares, comments) per post to determine the posts' efficacy. A total of 981 Facebook and 291 Twitter posts were initially mined using our keywords, and following filtration, 174 posts related to NHS audiology change of service were included for analysis. The results were then analysed geographically, along with an assessment of the interactions and sentiment analysis within the included posts. NHS Trusts and Boards should consider incorporating and promoting social media to communicate service changes. Users would be notified of service modifications in real-time, and different modalities could be used (e.g. videos), resulting in a more efficient service.

## Introduction

To mitigate the exponential spread of Covid-19 cases, the government took quick and decisive steps that affected healthcare services, including audiology departments in the UK. Consequently, there was a transition from in-person to teleaudiology patient care, which resulted in

com/kiadashtipour/Sentiment-Analysis-Adeel-Hussain].

**Funding:** This research is supported by the UK EPSRC COG-MHEAR programme (Grant No. 260 EP/M026981/1).

**Competing interests:** The authors have declared that no competing interests exist.

the cancellation of all in-person appointments for routine services. Moreover, new assessments and existing hearing aid users were no longer able to access their regular care pathways in the event of hearing difficulties [1, 2]. While infrastructure for remote audiology services had begun implementation prior to the Covid-19 pandemic, teleaudiology pathways were not yet fully established nor widely adopted in clinical practice.

Prior to the service being shifted to remote or postal services, audiology was predominantly a face-to-face facility for patients. There are several reasons for not adopting telehealth, including the absence of teleaudiology infrastructure, certain procedures necessitating face-to-face involvement (such as otoscopy), the necessity for sound-treated rooms for testing, and the multiple face-to-face appointments required for hearing aid fitting, counselling, and trouble-shooting. Previous research [2, 3] has also demonstrated reticence towards teleaudiology being used in clinical settings. Eikelboom and Swanepoel [3], surveyed 269 audiologists from around the world and found that only 15% had used teleaudiology, despite being confident in using the required technology [3]. However, despite this initial reticence teleaudiology had to be increasingly utilised during national and regional lockdowns in the COVID-19 pandemic in the United Kingdom. Saunders and Roughley surveyed 120 UK-based audiologists during the Covid-19 pandemic and found that while 98% of participants used teleaudiology at the time of the study, only 30% had used it beforehand. Participants responded positively to the remote pathways, but also highlighted the need for improvements in training and infrastructure as well as the potential to impact negatively on personal interactions [4].

The ways in which clinical services communicate with patients is important in an age where patients are increasingly seeking more information and encouraged to take responsibility for managing their health. Previously, individuals with a disability or health issue would seek advice and assistance from their friends, family, or healthcare experts. However, with improved internet accessibility and social media, individuals have more autonomy in gathering information from a wide range of resources [5]. This trend was exacerbated during the Covid-19 pandemic where face-to-face communication with healthcare providers decreased. In April 2020, research studies showed that people in the United Kingdom spent an average of four hours a day online, up from three hours and 29 minutes in September 2019, further supporting the aforementioned [6]. Specifically among adults aged 59 or older, the most likely demographic to utilise audiology services, studies have demonstrated that Facebook and Twitter to be the most commonly used social media platforms [7]. Furthermore analysis of social media usage across different platforms revealed variation in usage patterns among adults aged 59 years and older. The data showed that Facebook and Twitter were the most frequently used sites by this older demographic, with 40.98% and 17.20% of respondents aged 59+ reporting that they accessed these platforms several times per day, respectively. Additionally, 23.71% and 13.98% of those aged 59+ stated that they used Facebook and Twitter everyday in comparison with other platforms such as YouTube, Instagram, and Snapchat which were accessed less frequently [8]. Social media has also been utilised previously for understanding public perceptions, information dissemination and updates during public health crises [9–11].

Approximately 430 million people worldwide and 11 million individuals in the United Kingdom are estimated to have some degree of hearing loss, approximately, 1 in 6 of them having severe or profound hearing loss [12]. A listener's level of difficulty will vary based on the type of hearing loss they have, the environment they are in, whether their hearing loss has been treated, and how effectively their hearing aids are functioning. Since face-to-face communication was restricted due to the lockdown, audiology services increasingly turned towards social media to update patients on changes in their services. However, it is unclear how many services utilised social media to maintain communication with patients and how this was received by patients.

Our study is the first of its kind to investigate the role and impact of social media in engaging audiology consumers on service changes during the initial 6 weeks of the Covid-19 pandemic. The initial 6 week period was selected to ascertain the initial use of social media platforms. If the findings reveal that social media was not employed during this period, the study can provide valuable insights for future implementation. Specifically, we aim to analyse how audiology departments across health trusts in the United Kingdom utilised Facebook and Twitter, to communicate with their patients.

## Materials and methods

### Ethics

Due to the availability of the data used in this study in the public domain, no National Health Service (NHS) ethics review was required. As per our previous research and to ensure compliance with pertinent provisions of the General Data Protection Regulation (GDPR) [13–15], a comprehensive assessment was conducted to confirm that our study presented no privacy risk to individuals. We aimed to follow best practices for user privacy by excluding private information from our dataset. In addition, to comply with privacy laws and social media policies in accordance with the GDPR, to collect data, we did not disclose or publish direct posts by individuals, quotations from individuals, or the names or locations of users who are not public organisations or entities on the Facebook CrowdTangle platform and Twitter Application Programming Interface (API) [16].

### Data sources

We opted to use data from Facebook and Twitter, since they are the commonly used social media platforms [7, 8]. We utilised CrowdTangle in order to extract data from Facebook. CrowdTangle serves as a social media analytics tool, providing users with the capability to observe, analyse, and track the impact of content across various social media platforms. In addition, we leveraged the Twitter API to extract data from Twitter. The Twitter API facilitates access to Twitter's data [17], allowing for tasks such as extracting tweets, retrieving user information, and executing various other actions on the platform.

Within the period from March 23 to April 30, 2020, we specifically targeted English-language Facebook posts from pages in the UK. This particular time frame was selected due to its alignment with the initial announcement of the first UK lockdown [13]. In order to identify which audiology departments were utilising social media to communicate service changes, we narrowed our search to only include the initial six-week period of the lockdown.

The search terms used to extract data were hearing loss, hearing, difficulty, presbycusis, tinnitus, deafness, speech impairment, hearing aids, audiology, ear wax, ear syringing, microsuction, telecare, teleaudiology remote consultations. The comprehensive range of our search terms ensured that all relevant results were included during the searches.

Our initial search yielded 981 Facebook posts and 291 Twitter posts, which were then filtered to exclude posts unrelated to the NHS audiology departments. Our team manually filtered each post for messages of service change from NHS Trusts or partners, such as libraries where repair/battery clinics had previously been held. The final dataset did not include posts about private hearing clinics, other hearing-related news, or information from news outlets. This study also omitted information from the private sector and hearing aid manufacturers.

## Data analysis

CrowdTangle and Twitter API was used to calculate the frequencies of the various interactions from the filtered dataset. For the parameters of interest, specifically the number of posts and likes, data was collected and computed. Subsequently, these data points were visually represented using a heat map and circular chart. The presented heat map depicts the diverse geographical origins of the posts, with the colour and size of the circles serving as indicators of the quantity of interactions. To enhance the depth of analysis, sentiment analysis was applied to user comments accompanying the original posts [18].

In conducting sentiment analysis for Facebook messages, we employed a computational approach leveraging the TextBlob natural language processing library. TextBlob, a robust tool for sentiment analysis, utilises an algorithm to assign sentiment polarity scores. These scores are numerical representations indicating the sentiment polarity of the given text. In this context, TextBlob generates scores of -1, 0, and 1, corresponding to negative, neutral, and positive sentiments, respectively. The assigned scores function as quantitative indicators of the emotional content within the messages. A score of -1 denotes predominantly negative sentiment, +1 signifies positive sentiment, and 0 reflects a neutral message.

This methodological approach offers a systematic and objective framework for evaluating the sentiment expressed in Facebook and Twitter messages [19]. It contributes to a more nuanced understanding of sentiment dynamics within the textual content of social media interactions. The sentiment results were visually presented in the results section using a pie chart.

## Results

In this study, we investigated how NHS audiology departments utilised social media to provide service updates to patients. Fig 1 illustrates that most posts were generated between the third and fifth weeks.

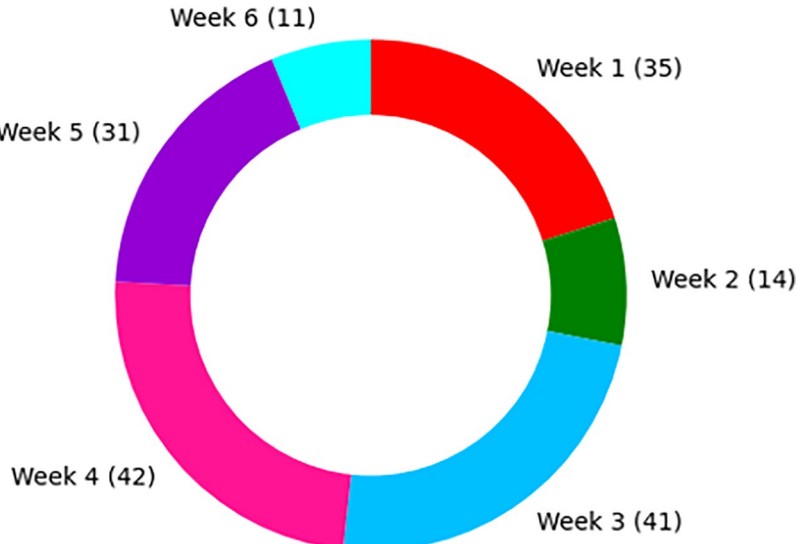

**Fig 1. Number of weekly posts related to audiology service changes over the 6-week lockdown period.** Colored segments represent the quantity of Facebook and Twitter posts extracted each week. Increased number of posts are indicated by larger segment sizes.

**Table 1. Number of shares, likes and comments on posts in England, Scotland, Wales and N. Ireland.**

| Country | Likes | Shares | Comments | Total |
|---|---|---|---|---|
| England | 915 | 856 | 236 | 2007 |
| Scotland | 415 | 737 | 323 | 1475 |
| Wales | 84 | 161 | 29 | 274 |
| Northern Ireland | 66 | 14 | 4 | 84 |

In our investigation of digital communication practices within healthcare organizations, we observed significant variations across regions. Among the 14 NHS Boards in Scotland, seven reported changes in service information through Facebook. Likewise, among the 217 NHS trusts in England, 41 conveyed changes in service information on both Facebook and Twitter. In Wales, three out of seven trusts reported changes in service information, while in Northern Ireland, only one out of five trusts reported a change in service information.

There was a total of 3,646 interactions across 174 posts, Table 1 illustrates the responses and interactions from the different countries across the UK. To be noted, this sample represents only a fraction of 11 million hearing aid users in the UK [12], some of the interactions may not have been from directly from the hearing aid user but possibly a friend or family member who may be affected by this change of service. Interactions included likes, shares, comments, or reposts. However, the reported interaction frequencies may underestimate the full reach, as some individuals may have viewed the posts without actively engaging through likes/comments.

Fig 2 shows a visual representation of the geographical distribution and number of interactions of Facebook and Twitter posts.] The heat/bubble map indicates that areas with a higher number of responses corresponds to areas with a higher number of interactions. In addition, the number of responses from the areas that posted service change updates in the U.K can be visualised. Both the gradient of the filled circle and the size of the circle indicate the number of posts and the number of interactions per post using the Virdis scale (blue to yellow and small to large) showing an increasing number of interactions.

Fig 3 visually represents the sentiment polarity derived from comments on the original posts. Each original post underwent a filtering process to gather comments from users. Comments where users tagged friends and family without additional written comments were excluded. The remaining comments were then analyzed using TextBlob, to gain an insight into sentiment of users towards social media posts relating to audiology service changes.

The overall sentiment within the comments was predominantly neutral (51%), with a positive sentiment accounting for 41%, and a minor portion (6%) expressing negativity. The sentiment trend exhibited stability, aligning with the sentiments mined. This suggests that social media updates on service changes were largely well-received. Some instances of negative sentiments reflected frustrations from individuals who lacked access to services. They expressed dissatisfaction with the performance of their hearing aids or devices, adding to their distress amid the lockdown and isolation during this uncertain period.

## Discussion

This study delved into the utilisation of social media platforms, specifically Facebook and Twitter, to disseminate information regarding audiology service changes within the NHS during an initial six-week period in the UK characterised by heightened uncertainty. The findings reveal that, in comparison to the total number of trusts and boards across the UK, the utilisation of social media to update patients was not optimised to its full potential.

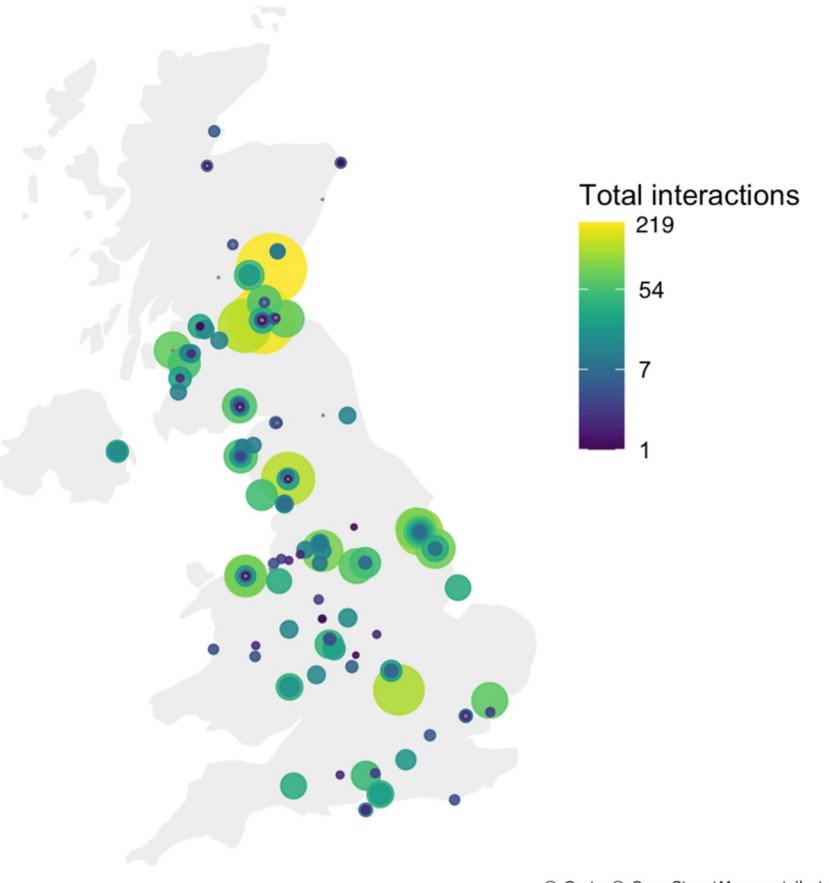

**Fig 2. Heat/bubble map of interactions with posts across the UK.** The color gradient (blue to yellow) and circle size indicate relative quantity of interactions in that location, with darker, larger circles representing more likes, comments, shares, etc.

Our analysis revealed a surge in the number of posts during the mid-weeks, with weeks 3 to 5 exhibiting the highest volume of posts. The sudden enforcement of lockdown measures, prompted by a significant increase in Covid-19 cases, may have led to a scenario where audiology departments were inundated with numerous tasks within a limited time frame. Consequently, some departments may not have immediately prioritised social media updates as they grappled with the urgent demands imposed by the evolving situation. It has been shown that individuals who have hearing loss can benefit from utilising electronic media as it improves their communication abilities and reduces auditory barriers [20]. Therefore, internet usage appeals to those who prefer text-based communication [21]. Our study identifies a low utilisation of social media platforms to communicate with patients and identifies the need for audiology services to leverage social media platforms to communicated service change updates.

Furthermore, the sentiment analysis conducted on the public's comments on the original posts indicates a predominant positive sentiment, with only a small percentage expressing a negative sentiment. This observation highlights the positive impact these posts had on the general public they reached. Additionally, Fig 2 illustrates regional disparities in the utilization of social platforms, and that significant portions of the map reflect few to no interactions, highlighting gaps in uptake of social media for patient communications during this time. In

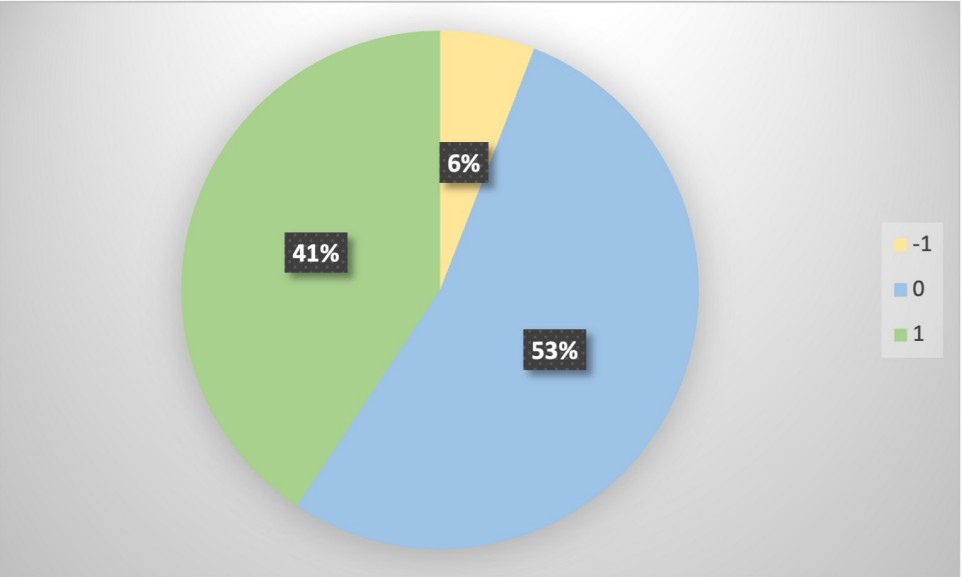

**Fig 3. Pie chart illustrating sentiment values extracted from user comments to service change posts.** Green represents positive sentiment (41%), blue represents neutral sentiment (51%) and yellow for negative sentiment (6%).

Scotland, boards in the southern part of the country demonstrated more active engagement compared to the Highlands, where posts were relatively scarce. Within England, the Midlands and northern regions exhibited a higher level of utilisation of social media compared to other parts of the country.

The current study provides evidence that sharing service change messages on reliable online platforms can effectively inform patients. However, the credibility of information shared on social media can be undermined by the circulation of false or misleading content, as indicated by previous research [7]. The failure to timely post accurate information on social media may result from the ambiguity surrounding lockdown protocols and a shortage of personnel due to the government furlough programme, which was initiated on 27 March 2020.

## Strengths and limitations

Our study has multiple strengths. It is the first study to assess communication by audiology departments on social media regarding service changes during the COVID-19 pandemic. We do this by utilising the two most frequently used social media platforms in older adults, namely Facebook and Twitter. Within these platforms we also utilised a comprehensive range of search terms in order to identify all social media posts from audiology centres across the entirety of the United Kingdom. We also utilise sentiment analysis, based on established methodology in this field of study [19], in order to better gain insight into service users perspective on department communication. This allows for easier comparison and analysis as the sentiment of a post can be accurately reflected as an objective metric.

There are also some limitations to our study. Our analysis is limited by our small sample size due to the short time period over which our study is conducted. This was primarily due to issues with data access as well as resource constraints. Whilst our study utilised the naive bayes classifier from the TextBlob package other approaches utilising modern deep learning architectures may have improved the accuracy of our sentiment analysis. Furthermore, although we

captured a significant proportion of communication on social media platforms we were unable to access other modes of communication with patients such as emails, SMS, and letters due to NHS data protection constraints. This means that we are unable to evaluate the communication from UK audiology centres beyond their social media platforms. Lastly, it is important to note that the demographic profile of patients engaging with these social media platforms likely diverges from that of the average service user, with a propensity towards a younger age bracket.

## Future directions

This is the first study of its kind that sought to assess the use of social media platforms for monitoring communication related to changes in audiology services. In future work, we plan to extend our investigation by incorporating additional platforms, such as YouTube and other hearing aid forums, thereby enabling a more comprehensive content analysis that can be correlated with our overarching findings. To deepen our understanding of the communication strategies adopted by healthcare providers, we may consider supplementing social media-mined data with surveys distributed to various NHS trusts. These surveys will help discern the approaches employed by these trusts in notifying patients of service changes, offering insights for potential enhancements to deliver an enhanced service.

Moreover, we would also aim to analyse and compare traditional communication channels, such as emails, short messaging service, and letters, with social media updates. Conducting patient surveys will allow us to inquire about their preferred communication methods, enabling us to determine what suits both patients and clinics best. Such an approach will enable a cost-benefit analysis of traditional methods versus social media, guiding more effective allocation of resources to improve overall service and message delivery.

While our current study focused on a short initial six-week period, we aspire to extend our investigation to understand how social media was utilised before, during, and after the Covid-19 pandemic. The scope of the study, previously limited to audiology services, may be broadened to encompass cross-departmental studies, including the role of primary care. This expansion would provide insights into how other departments and sectors, such as the pharmaceutical industry, have leveraged social media, identify potential benefits, and guide strategies for enhancing audiology services based on successful practices in different healthcare domains.

## Conclusion

To conclude, we sought to investigate how NHS audiology departments used social media to notify patients of service changes. This preliminary first of its kind study has shown that the potential of social media is significantly under utilised in audiology research and practice, as supported by the lack of posts or information on service changes on a number of NHS trusts' social media platforms. This highlights the need for audiology healthcare practitioners who work with patients with hearing loss to be aware of social media as a potential platform for dissemination of information among their patient population. A well-informed workforce would provide more effective service and would in turn lead to greater patient satisfaction and improved patient outcomes.

## Author Contributions

**Conceptualization:** Adeel Hussain, Zain Hussain, Aziz Sheikh, Amir Hussain.

**Data curation:** Adeel Hussain.

**Formal analysis:** Adeel Hussain, Dominic Ng, Muhammed Shaan Riaz.

**Investigation:** Adeel Hussain, Zain Hussain.

**Methodology:** Adeel Hussain, Zain Hussain, Dominic Ng.

**Software:** Adeel Hussain.

**Supervision:** Aziz Sheikh, Amir Hussain.

**Writing – original draft:** Adeel Hussain.

**Writing – review & editing:** Adeel Hussain, Zain Hussain, Mandar Gogate, Kia Dashtipour, Dominic Ng, Muhammed Shaan Riaz, Adele Goman, Aziz Sheikh, Amir Hussain.

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
