## [Decision Letter · Decision Letter 0]

17 Aug 2023

PONE-D-23-19309Impact of the Covid-19 Pandemic on Audiology Service Delivery: Observational Study of the Role of Social Media in Patient CommunicationPLOS ONE

Dear Dr. Hussain,

Thank you for submitting your manuscript to PLOS ONE. After careful consideration, we feel that it has merit but does not fully meet PLOS ONE’s publication criteria as it currently stands. Therefore, we invite you to submit a revised version of the manuscript that addresses the points raised during the review process.

The two reviewers have suggestions and the one has suggestions about how to analyze you data in more detail. I have added more specific details in my comments in the form. It is important for acceptance that you address all of the suggestions from the three of us. Personally I find the information important but feel that it needs more detail and that your data has not yet finished telling you its story.

We look forward to receiving your revised manuscript.

Kind regards,

Mary Diane Clark, PhD

Academic Editor

PLOS ONE

Journal Requirements:

   "This research is supported by the UK EPSRC COG-MHEAR programme (Grant No.EP/M026981/1)."

4. We note that Figure 2 in your submission contain [map/satellite] images which may be copyrighted. All PLOS content is published under the Creative Commons Attribution License (CC BY 4.0), which means that the manuscript, images, and Supporting Information files will be freely available online, and any third party is permitted to access, download, copy, distribute, and use these materials in any way, even commercially, with proper attribution. For these reasons, we cannot publish previously copyrighted maps or satellite images created using proprietary data, such as Google software (Google Maps, Street View, and Earth). For more information, see our copyright guidelines: http://journals.plos.org/plosone/s/licenses-and-copyright.

Additional Editor Comments:

Thanks for submitting your paper to Plos One. It is an important topic and these impacts will most likely have long-term impacts. I have some additional concerns that are not mentioned by the two reviewers. They are listed here.

1--Please spell out NHS the first time. I know in the UK it is well known but it took me a minute to get it.

2--On line 22, please start a new paragraph

3--line 32 has a change in topic so please be sure your paragraph is internally consistent

4--Your introduction is extremely brief. COVID was not the first time that this type of service was online. I checked and there is a long history of moving towards this type of service. Please expand the introduction.

5--CrowdTangle (notice the Capital T) is not common knowledge. Therefore, you are going to need to explain that system in more detail. Also you have it spelled two different was":Crowd Tangle and Crowdtangle. Please use the correct name.

6--please define API

7--It is not clear what parameters you are targeting. Please expand on exactly what are your parameters of interest.

8--Please separate the results and discussion

9--you need a better key to explain Figure 1. I still am not clear what the colors mean

10--Table 1 is redundant to the text so please delete it. Rule is either a table or in the text--one or the other

11--You need more interpretation of your results

12--Table 2 needs to be expanded for clarity.

Reviewers' comments:

Reviewer's Responses to Questions

**Comments to the Author**

1. Is the manuscript technically sound, and do the data support the conclusions?

Reviewer #1: Yes

Reviewer #2: Yes

2. Has the statistical analysis been performed appropriately and rigorously? 

Reviewer #1: N/A

Reviewer #2: Yes

3. Have the authors made all data underlying the findings in their manuscript fully available?

Reviewer #1: Yes

Reviewer #2: Yes

4. Is the manuscript presented in an intelligible fashion and written in standard English?

Reviewer #1: Yes

Reviewer #2: Yes

5. Review Comments to the Author

Reviewer #1: Reviewer comments re: "Impact of the Covid-19 Pandemic on Audiology Service Delivery: Observational Study of the Role of Social Media in Patient Communication"

See article comment bubbles for grammar and related comments.

The methodology employed in the study appears to be sound for the specific research question and objectives stated in the paper. The researchers aimed to investigate how NHS audiology departments in the UK utilized social media to communicate service changes during the Covid-19 pandemic. To achieve this goal, they collected data from Facebook and Twitter, focusing on a specific time frame (March 23 to April 30, 2020) that coincided with the initial announcement of the first UK lockdown.

The use of data extraction tools like Crowd tangle and Twitter API for collecting relevant posts aligns with the objective of analyzing social media communication. The researchers applied specific search terms to ensure comprehensive results and filtered the data manually to include only posts directly linked to NHS audiology service communication. Moreover, the study considered ethical considerations and data privacy laws, as the information was gathered from the public domain.

The geospatial mapping and interaction analysis were appropriate methods for analyzing the geographic distribution of posts and understanding user engagement with the content. The study also acknowledged its limitations, such as the inability to access private communications, which may have impacted the comprehensive understanding of all communication strategies.

Considering the research question and objectives, the methodology used allowed the researchers to gain insights into how NHS audiology departments used social media to communicate service changes during the pandemic. However, it is essential to recognize that the study's scope is specific to social media usage during a particular period and may not address all aspects of audiology service changes comprehensively.

SUGGESTIONS - To further enhance the study's depth and gain additional insights, the researchers could have considered conducting the following analyses:

Content Analysis: Instead of solely focusing on the number of interactions (likes, shares, comments), the researchers could have performed a content analysis of the posts. This analysis would involve categorizing the posts based on the type of information shared (e.g., changes in appointments, teleaudiology services, hearing aid maintenance), the tone of the messages (e.g., informative, reassuring, urgent), and the overall effectiveness of the communication.

Sentiment Analysis: By employing sentiment analysis, the researchers could assess the emotional tone of the social media posts. This analysis would help determine whether the messages conveyed by the audiology departments were perceived positively, negatively, or neutrally by the audience. Sentiment analysis can provide valuable insights into how well-received the service changes were among patients.

Comparison with Traditional Communication Channels: In addition to analyzing social media usage, the researchers could have compared the effectiveness of social media as a communication channel with more traditional methods like emails, SMS, or letters. By surveying patients who received information through different channels, the researchers could gauge the preferences and efficacy of various communication methods.

Longitudinal Analysis: The study focused on a specific six-week period during the initial announcement of the UK lockdown. A longitudinal analysis covering multiple time periods (e.g., before, during, and after the lockdown) could reveal how communication strategies evolved over time and how patients' responses changed as the pandemic progressed.

Qualitative Interviews or Surveys: To gain a more comprehensive understanding of patient perspectives and experiences, the researchers could have conducted qualitative interviews or surveys with patients who interacted with the audiology departments' social media posts. These interviews could provide valuable insights into patients' satisfaction, comprehension of information, and suggestions for improvement.

Comparison with Other Healthcare Specialties: The study focused on audiology service changes, but a broader analysis comparing social media communication practices across various healthcare specialties could highlight similarities, differences, and best practices for effective communication during a public health crisis.

Analysis of Responses to Misinformation: Since the study touched upon concerns about the credibility of information shared on social media, the researchers could have analyzed responses to misinformation related to audiology services during the pandemic. This analysis could help identify potential challenges in combating false or misleading content.

Cost-Benefit Analysis: A cost-benefit analysis of using social media as a communication channel could provide insights into the resource allocation for social media efforts compared to other communication methods. This analysis would be useful in determining the most cost-effective ways to reach and engage with patients.

By incorporating these additional analyses, the researchers could have gained a more comprehensive understanding of the role and impact of social media in engaging audiology consumers during the Covid-19 pandemic and beyond.

Reviewer #2: p. 2, lines 32-33: You cite a study concluding that Facebook and Twitter are the most commonly used types of social media. Was any effort made to determine the types of social media most commonly used by the ages and other demographics of people who tend to use audiology services more regularly than others? Given that older adults are one of the largest groups seen by audiologists and that "presbycusis" was one of your search terms, I'm wondering whether your choices of Facebook and Twitter were based at all on older adult social media use patterns.

Also on the topic of populations that use audiology services: the search terms that you list seem primarily focused on adult audiology services. Were pediatric audiology services included in the social media posts that you analyzed? Or was this study primarily about adult audiology services?

p. 3, lines 94-95: Is there a reason to believe that only hearing aid users are interacting with these posts and therefore represented in the total number of interactions? The meaning of the sentence here isn't entirely clear to me. Are you trying to point out that not all hearing aid users are being reached by social media posts? Something else?

p. 4, lines 110-111: Why is YouTube coming up this far into the paper when Facebook and Twitter were the platforms used for collecting data for the study? And if 40% of participants reported using Facebook and YouTube in the study you cite at this point in the paper, is there a reason that you didn't use YouTube as a data source? I see that later in the paper, you mention YouTube as a future direction. Still, the mention of YouTube in the Results and Discussion section is somewhat confusing.

This paper addresses an important topic for audiology services. The transition from in-person to virtual services at the beginning of the pandemic was jarring for many audiologists around the world, and understanding how different health systems used social media to keep patients informed will be helpful for improving how we use social media in audiology.

6. PLOS authors have the option to publish the peer review history of their article (what does this mean?). If published, this will include your full peer review and any attached files.

Reviewer #1: No

Reviewer #2: No

---

## [Author Response · Author response to Decision Letter 0]

23 Feb 2024

Response to Editor and Reviewer Comments

We would like to thank the Editor and Reviewers for their insightful comments on our study. We believe these have improved our study, and have provided point-by-point responses below. 

Editor Comments:

1--Please spell out NHS the first time. I know in the UK it is well known but it took me a minute to get it.

Response: We have now spelled out as follows, “National Health Service (NHS)” , and updated this across the paper. 

2–On line 22, please start a new paragraph

Response: We have amended accordingly.

3–line 32 has a change in topic so please be sure your paragraph is internally consistent

Response: We have amended this accordingly, and separated paragraphs.

4--Your introduction is extremely brief. COVID was not the first time that this type of service was online. I checked and there is a long history of moving towards this type of service. Please expand the introduction.

Response: We agree with respectable reviewer, we have added more information to the introduction clarifying that Teleaudiology had been looked into prior to Covid but still lacked infrastructure.

5--CrowdTangle (notice the Capital T) is not common knowledge. Therefore, you are going to need to explain that system in more detail. Also you have it spelled two different was":Crowd Tangle and Crowdtangle. Please use the correct name.

Response: Thank you for this comment, we have corrected CrowdTangle spelling within report, we have also detailed what CrowdTangle is capable of for a better understanding of the reader.

6--please define API

Response: Thanks for the reviewer for highlighting this, it has now been defined appropriately within the script, Application Programming interface (API), we have also detailed what the API is capable of doing for a better understanding for the reader. 

7--It is not clear what parameters you are targeting. Please expand on exactly what are your parameters of interest.

Response: We agree with respectable reviewer, we have clarified the parameters further so that they are understandable to the reader clearly

8--Please separate the results and discussion

Response: Thank you for this, we have now separated the results and discussion section. 

9--you need a better key to explain Figure 1. I still am not clear what the colors mean

Response: We have explained this to further help clarify what the different colours are illustrating.

10--Table 1 is redundant to the text so please delete it. Rule is either a table or in the text--one or the other

Response: Thank you, we have removed Table 1. 

11--You need more interpretation of your results

Response: We agree with this and have expanded our results and discussion sections to provide more detail. 

12--Table 2 needs to be expanded for clarity.

Response: We agree with this, and have explained the table more clearly in the results.

Reviewer 1 Comments:

To further enhance the study's depth and gain additional insights, the researchers could have considered conducting the following analyses:

Content Analysis: Instead of solely focusing on the number of interactions (likes, shares, comments), the researchers could have performed a content analysis of the posts. This analysis would involve categorizing the posts based on the type of information shared (e.g., changes in appointments, teleaudiology services, hearing aid maintenance), the tone of the messages (e.g., informative, reassuring, urgent), and the overall effectiveness of the communication.

Response: We agree with the reviewer that content analysis would have been useful for further insights, however on our initial reading/scanning of interactions and comments on Facebook, we found these to be largely limited to short replies and ‘reactions’ to posts. It would be useful for us to analyse other social media sources, e.g. hearing aid user forums, twitter etc, for content analysis and correlate them to our high-level findings. We have now made reference to this in our future work and limitations section. We have however carried out sentiment analysis on the available comments, as discussed in the next response.

Sentiment Analysis: By employing sentiment analysis, the researchers could assess the emotional tone of the social media posts. This analysis would help determine whether the messages conveyed by the audiology departments were perceived positively, negatively, or neutrally by the audience. Sentiment analysis can provide valuable insights into how well-received the service changes were among patients.

Response: Thank you for this insightful comment. We agree it would be useful to understand public sentiment towards these posts. As noted earlier, most of the patient engagement was limited to short comments or ‘reactions’ to posts with different emotions. However, we have extracted available comments and performed high level sentiment analysis on them , to gain some insight into public sentiment towards the service communications. 

Comparison with Traditional Communication Channels: In addition to analyzing social media usage, the researchers could have compared the effectiveness of social media as a communication channel with more traditional methods like emails, SMS, or letters. By surveying patients who received information through different channels, the researchers could gauge the preferences and efficacy of various communication methods.

Response: This is an excellent suggestion, however we feel a larger experiment would be required where we obtain consent from patients enrolled to access their other communications. We have made reference to this in our future work.

Longitudinal Analysis: The study focused on a specific six-week period during the initial announcement of the UK lockdown. A longitudinal analysis covering multiple time periods (e.g., before, during, and after the lockdown) could reveal how communication strategies evolved over time and how patients' responses changed as the pandemic progressed.

Response: This is an excellent suggestion, however we feel a larger experiment would be required where we obtain consent from patients enrolled to access their other communications. We have made reference to this in our future work.

 Qualitative Interviews or Surveys: To gain a more comprehensive understanding of patient perspectives and experiences, the researchers could have conducted qualitative interviews or surveys with patients who interacted with the audiology departments' social media posts. These interviews could provide valuable insights into patients' satisfaction, comprehension of information, and suggestions for improvement.

Response: This is an excellent suggestion, however we feel a larger experiment would be required where we obtain consent from patients enrolled to access their other communications. We have made reference to this in our future work.

Comparison with Other Healthcare Specialties: The study focused on audiology service changes, but a broader analysis comparing social media communication practices across various healthcare specialties could highlight similarities, differences, and best practices for effective communication during a public health crisis.

Response: This is an excellent suggestion, however we feel a larger experiment would be required where we obtain consent from patients enrolled to access their other communications. We have made reference to this in our future work.

Analysis of Responses to Misinformation: Since the study touched upon concerns about the credibility of information shared on social media, the researchers could have analyzed responses to misinformation related to audiology services during the pandemic. This analysis could help identify potential challenges in combating false or misleading content.

Response: This is an excellent suggestion, however we feel a larger experiment would be required where we obtain consent from patients enrolled to access their other communications. We have made reference to this in our future work.

Cost-Benefit Analysis: A cost-benefit analysis of using social media as a communication channel could provide insights into the resource allocation for social media efforts compared to other communication methods. This analysis would be useful in determining the most cost-effective ways to reach and engage with patients.

Response: This is another excellent suggestion, however we feel a larger experiment would be required where we obtain consent from patients enrolled to access their other communications. We have made reference to this in our future work. 

Reviewer 2 comments:

p. 2, lines 32-33: You cite a study concluding that Facebook and Twitter are the most commonly used types of social media. Was any effort made to determine the types of social media most commonly used by the ages and other demographics of people who tend to use audiology services more regularly than others? Given that older adults are one of the largest groups seen by audiologists and that "presbycusis" was one of your search terms, I'm wondering whether your choices of Facebook and Twitter were based at all on older adult social media use patterns.

Response:Thank you for this comment from the reviewer. To further support the use of Facebook and Twitter within this study another study has been cited were the authors had found that the use of facebook and twitter was larger in the older population in comparison to other social platforms.

Also on the topic of populations that use audiology services: the search terms that you list seem primarily focused on adult audiology services. Were pediatric audiology services included in the social media posts that you analyzed? Or was this study primarily about adult audiology services?

Response: Thank you for the comment from the reviewer, all terminology used within the search terms would have been appropriate for both adult and children services except “Presbycusis” which refers to age related hearing loss. 

p. 3, lines 94-95: Is there a reason to believe that only hearing aid users are interacting with these posts and therefore represented in the total number of interactions? The meaning of the sentence here isn't entirely clear to me. Are you trying to point out that not all hearing aid users are being reached by social media posts? Something else?

Response: Many thanks for the reviewer for this comment, we have clarified the sentence so that it is more understandable, “some of the interactions may not have been from directly from the hearing aid user but possibly a friend or family member who may be affected by this change of service. Interactions included likes, shares, comments, or reposts. However, the reported interaction frequencies may underestimate the full reach, as some individuals may have viewed the posts without actively engaging through likes/comments.” 

p. 4, lines 110-111: Why is YouTube coming up this far into the paper when Facebook and Twitter were the platforms used for collecting data for the study? And if 40% of participants reported using Facebook and YouTube in the study you cite at this point in the paper, is there a reason that you didn't use YouTube as a data source? I see that later in the paper, you mention YouTube as a future direction. Still, the mention of YouTube in the Results and Discussion section is somewhat confusing.

Response: Thank you to the reviewer for this comment. We have now ensured that the mention of specific social media platforms is now clearer and in a more appropriate order.

---

## [Editor Report · Decision Letter 1]

4 Mar 2024

Impact of the Covid-19 Pandemic on Audiology Service Delivery: Observational Study of the Role of Social Media in Patient Communication

PONE-D-23-19309R1

Dear Dr. Hussain,

We’re pleased to inform you that your manuscript has been judged scientifically suitable for publication and will be formally accepted for publication once it meets all outstanding technical requirements.

Kind regards,

Mary Diane Clark, PhD

Academic Editor

PLOS ONE

Additional Editor Comments (optional):

Thank you for your work on this revision. I enjoyed reading this revision and believe it will make an important contribution. I have recommend acceptance.